# A New Methodology for Rockfall Hazard Assessment in Rocky Slopes

**Larissa Regina Costa Silveira** [1] , **Milene Sabino Lana** [2,*], **Pedro Alameda-Hernández** [3] and **Tatiana Barreto dos Santos** [2]

1    Science and Technology of Materials Department, Federal University of Bahia, Salvador 40170-110, BA, Brazil
2    Mining Engineering Department, Federal University of Ouro Preto, Ouro Preto 35400-000, CEP, Brazil
3    Urban Engineering Department, Federal University of Ouro Preto, Ouro Preto 35400-000, CEP, Brazil
*    Correspondence: milene@ufop.edu.br

**Abstract:** This article presents an approach to rockfall hazard assessment for rocky slopes based on a previously published rockfall hazard methodology. The original method is appropriate to high alpine rocky slopes exposed to large scale deformations. It evaluates the parameters related to the geomechanical characterization of rock mass, indications of activity, external influences and event intensity. The original methodology was modified to consider different contexts, including geological, climatic and social environments. Parameters related to external influences were modified; the geometry and characteristics of the slope and the catchment area were introduced. The original methodology and the new proposal were applied to two urban slopes and one railway slope in order to test and compare the methods. The original proposal could not represent the rockfall conditions of these slopes. The new proposal was validated using two mine slopes, whose conditions of stability are known. The results of the analyses with the urban slope and the railway slope were coherent with the situation observed at the field. The validation in the mine slopes showed that this approach is applicable in several situations, being able to determine how hazardous a slope is in relation to rockfall events.

**Keywords:** rockfall hazard system; probability matrix; hazard matrix; urban slopes; railway slopes; mine slopes

## 1. Introduction

Several systems of susceptibility, vulnerability, hazard and risk assessment have been proposed because they are easy-to-use and efficient tools for accident prevention and management. Researches on susceptibility, vulnerability, hazard and risk have been carried out in several science fields, such as geology and geotechnics, environmental contamination and ecology [1–7].

Rockfalls are hard to predict because the rock blocks usually do not present previous movement signs and they quickly fail. This condition worsens when monitoring measures are not available or when the rockfall hazard is neglected. Monitoring measures are not available especially in peripheral urban areas and, sometimes, in ecological or adventure tourism areas. A recent example of a serious accident involving a high magnitude rockfall occurred in January, 2022 in Capitólio city, a cliff region of Minas Gerais State of Brazil. In this case, a high quartzite rock block toppled and hit a boat with tourists in a lake located in this region. Another accident involving rockfall, also in Brazil, occurred in January, 2021 at a quarry in the metropolitan region of Salvador, State of Bahia, when a rock block fell from an operational slope onto an excavator, causing the death of one operator.

In populated mountainous regions, rockfalls constitute a major hazard once they can cause damage to properties and personal injuries. Therefore, a geotechnical hazard assessment in these areas is essential, and it consists of the first step of future mitigation

planning and risk management. Through geotechnical hazard assessment, it is possible to define the areas with the most urgency of intervention and with the need of control or mitigation measures. These measures can include a constant monitoring plan, block support with bolts and high-resistance screens and removal of overhanging blocks.

In view of the importance of rockfall hazard classification, the main objective of this paper is to propose an easy-to-use approach to rockfall hazard assessment. This approach was adapted from the methodology proposed by [8]. This methodology is easy to use, but it is only suitable for high urban slopes, in alpine regions. Thus, the methodology proposed in this research aims to improve the original proposal by the accurate description of the slope geometry, adaptation of the methodology to rainy regions and inclusion of other seismic situations, like mine blasting and heavy equipment traffic.

The original and the new proposal were applied in three case studies. The first studied slope is located in Mariana town; the second one is located on the railway which connects the towns of Ouro Preto and Mariana; finally, the third studied slope is located in Ouro Preto; all of them in Minas Gerais, Brazil. The locations of the studied slopes are shown in Figure 1. Furthermore, in order to validate the new proposal, the method was applied to two slopes in a quartzite mine, located in São Thomé das Letras town (320 km from Ouro Preto), Brazil (Figure 1), whose stability conditions regarding rockfalls are known.

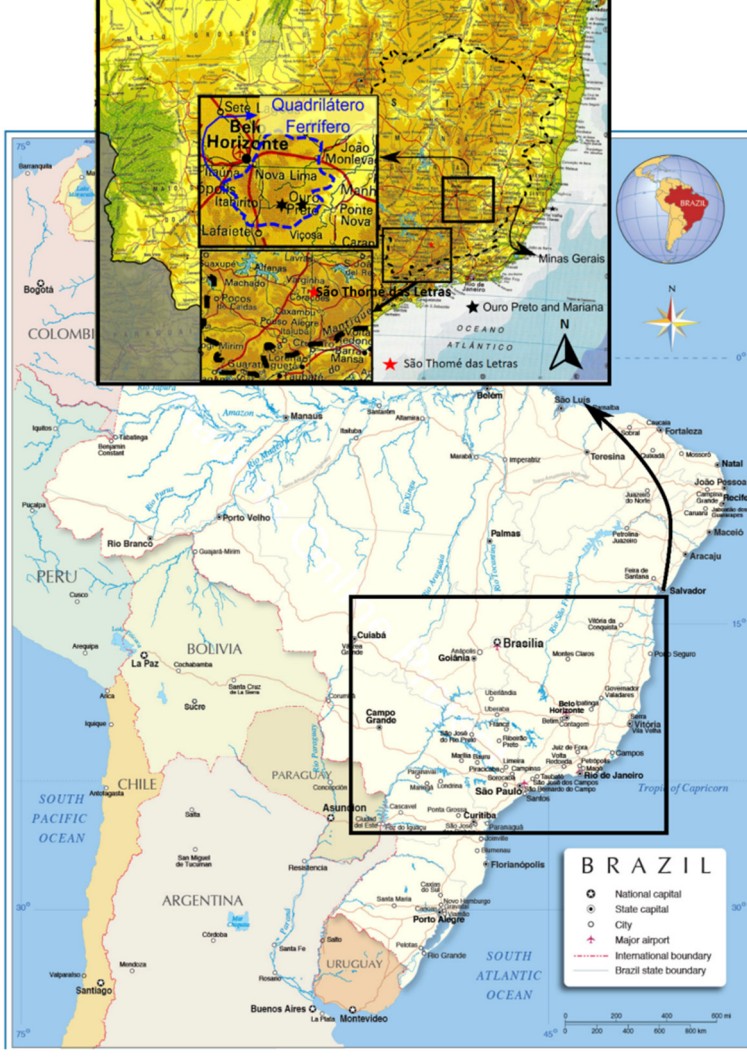

**Figure 1.** Location of the slopes in Minas Gerais state, Brazil (modified from nationsonline.org (accessed on 22 October 2022).

## 2. Rockfall Hazard and Risk Assessment Methodologies

### 2.1. Concepts

This article applies the concepts defined by the Technical Committee 32 from the International Society of Soil Mechanics and Geotechnical Engineering—ISSMGE [9]. These definitions are internationally accepted and, according to [9], should be used for all zoning, reports and land use planning documents in order to avoid misunderstanding of the terms:

- Susceptibility: a quantitative or qualitative assessment involving rock mass or soil classification, volume (or area) and spatial distribution of mass movement which exists or potentially may occur.
- Hazard: a potential condition that can lead to an undesirable consequence. The hazard description should include the location, volume (or area), classification, velocity of the potential mass movement and the probability of its occurrence within a given time.
- Risk: a measure of the probability of the event occurring and the consequences to health, property or the environment. It is mathematically defined by the multiplication of the failure probability and the consequences of this failure.

### 2.2. Rockfall Hazard Assessment Methodologies

Several authors proposed methodologies to access the rockfall hazard and risk conditions of slopes. Some of the main proposed methodologies are appropriate for highway slopes. These methodologies consider the traffic parameters, the structural condition and the geometry of the slopes, the catchment area and the previous instability or rockfall frequency. The Rockfall Hazard Rating System (RHRS), ref. [10] and the Modified Colorado Rockfall Hazard Rating System (CRHRS) [11] are examples of hazard assessment systems for highway slopes. RHRS does not establish a hazard (or risk, in the case of the CRHRS) classification, as "low, medium or high hazard/risk". The final result of these methodologies is an index that yields the most problematic regions, associated with the high score values.

Another system of rockfall hazard assessment focused on highway slopes was proposed by [12]. The first step of this approach generates a hazard index from a quick slope data collection. The second step generates an index from detailed field data.

Regarding urban areas, they can be cited a quantitative risk classification proposed by [13], in which risk is obtained by a risk matrix. This system was developed using slopes from a mountainous region in Norway. Nine parameters are used to describe the rockfall slope risk. These criteria can be organized into two groups: the first group is related to the structural conditions of the rock slope, and the second one is related to the displacement rates and activity indications. The consequence is determined by loss of lives. Although it is an effective method, it is limited to populated urban areas; and some parameters, especially those related to mass displacements, require constant monitoring of slopes, which is not always possible, especially in poorly peripheral urban areas.

According to [14], the Slope Mass Rating (SMR) [15] estimates the rockfall hazard. The SMR is calculated by adjustments to the Rock Mass Rating (RMR) [16], multiplying some factors to the basic RMR; for instance, a factor determined by the spatial position of the discontinuities in relation to the slope dip decreases the RMR value. Thus, the SMR classification is not a hazard classification, but a susceptibility index.

Another method focused on urban areas was proposed by [8]. This method evaluates the rockfall hazard for the Bavarian Alps, which takes into account the occurrence probability and the intensity of potential events. It is based on geological-geotechnical data collected in the field and observations. The rockfall probability is related to structural conditions of the slope; to the geomechanical environment, that considers parameters related to the rock mass displacement; and to the activity indications and the external influences, as precipitation and seismic zones. The intensity is related to the volume of failed material is due to the rockfall event. The method is easy-to-use, and it is applicable in urban areas located in mountainous regions.

The methodologies aforementioned are efficient, and some of them are widespread internationally, such as RHRS [10]. Furthermore, they are easy to use, which is an important feature in geotechnical hazard and risk analysis routines. However, they present some limitations regarding applicability. Some of them are suitable only for highway slopes, others only for urban slopes, precisely for alpine regions. Others do not present a hazard analysis. Thus, it is important to propose an easy-to-use methodology, as well as those cited, but one that is more flexible and able to be applied in different contexts. Thus, the described methods were used as a basis for this proposal, considering their strong points and disregarding their weaknesses.

## 3. Materials and Methods

The main objective of this research is to provide an appropriate methodology for rockfall hazard assessment of rock slopes from urban areas, mines, highways and railways, considering all relevant parameters. Among the methodologies found in the literature, the approach proposed by [8] was selected to be modified, forming the new proposal.

The method was selected because it is an easy-to-use tool based on parameters readily obtained in the field. In addition, the methodology is a preliminary system, which has not been thoroughly tried out and the authors recommended its application and encouraged adaptations. According to [8], adjustments have to be carried out regarding the graduation/scales of disposition and activity as well as of external impacts.

Three slopes susceptible to rockfalls were selected to carry out the rockfall hazard analysis. In case of rockfall occurrence in these slopes, serious consequences are expected, with material losses and damage to the physical integrity of the exposed people. These slopes are: Slope-1, located in the Cabanas neighborhood (Mariana, Minas Gerais District, Brazil); Slope-2, located on the railway which connects the towns of Ouro Preto and Mariana; and Slope-3, located in the Vila Aparecida neighborhood (Ouro Preto, Minas Gerais District, Brazil).

For all slopes, field inspections were made with the purpose of reconnaissance of the area and delimitation of access points to carry out a survey of geotechnical and local characteristics. It was possible to observe and survey both the parameters related to the original methodology developed by [8], as well as other parameters that apparently had an influence on the rockfall hazard in these places, but which were not taken into account in the original methodology of [8]. These steps made possible the proposition of a new system of rockfall hazard analysis, based on the proposal of [8]. In the following sections, the original methodology developed by [8] is presented, as well as the new proposal. Field observations and the application of both methods are presented in the results section.

### 3.1. Bauer and Neumann Original Method

The parameters used in [8] are presented in Tables 1 and 2. Rockfall susceptibility is evaluated by the disposition and the activity. Rock mechanics disposition parameters are discontinuity parameters (orientation, persistence, degree of transection, aperture, roughness and degree of loosening) and the weathering grade [17]. Geomechanical Environment disposition is evaluated by the type of basement, the large-scale deformations and the mass movement in the slope foot. The activity is evaluated by rockfall activity indicators. Weights are attributed to each parameter, and the susceptibility is obtained by their sum.

Following the susceptibility evaluation, the external impact is evaluated by the sum of the weights of precipitation and earthquakes. The rockfall probability matrix is presented in Figure 2a. It is the relationship between susceptibility and external impact. The Figure 2a output is the probability of rockfall occurrence; it is combined with the intensity of the event (consisting of the block volume, shown in Table 2) forming the hazard matrix (Figure 2b), finally obtaining the hazard assessment.

**Table 1.** Scores of the parameters and rockfall probability values, modified from [8].

| | | Parameter | Value | | |
|---|---|---|---|---|---|
| | | | Unfavorable | Fair | Favorable |
| Susceptibility (Disposition + activity) | Rock Mechanics (Disposition) | (1) Discontinuity orientation | 500 (Adverse or slope parallel) | 200 (horizontal) | 0 (Vertical or inward) |
| | | (2) Weathering | 200 (W4–W5) | - | 0 (W1–W3) |
| | | (3a) Discontinuity extent | 200 (>10 m) | 100 (1–10 m) | 0 (<1 m) |
| | | (3b) Degree of transection of discontinuities | 200 (No mineral bonds) | - | 0 (Mineral bonds existent) |
| | | (3c) Discontinuity aperture | 200 (>1 cm) | 100 (0.5–1 cm) | 0 (<0.5 cm) |
| | | (3d) Discontinuity roughness | 200 (Slickensides) | - | 0 (No slickensides) |
| | | (4) Degree of loosening (alternative to category 3) | 800 (Clear indications available) | 400 (Only subordinate indications) | 0 (No indications) |
| | Geomechanical Environment (Disposition) | (5) Type of basement | 600 (Dissolvable rocks) | 200 (Clayey—marly formations) | 0 (Other formations) |
| | | (6) Large-scale, deep-seated deformations | 200 (yes) | - | 0 (no) |
| | | (7) Mass movements in the slope foot | 200 (yes) | - | 0 (no) |
| | (8) Activity | Initial Activity | 500 (Active) | - | 0 (Not active) |
| | Total Susceptibility (1 + 2 + (3 or 4) + 5 + 6 + 7 + 8) | | | | |
| External Impact | | (9) Precipitation | 50 (>310 mm) | 20 (230–310 mm) | 0 (<230 mm) |
| | | (10) Earthquakes | 50 (Zone 2) | 20 (Zone 1) | 0 (Zone 0) |
| | Total External Impact (9 + 10) | | | | |

**Table 2.** Rockfall intensity, modified from [8].

| Intensity | High Magnitude Rockfalls | | | Low Magnitude Rockfalls | |
|---|---|---|---|---|---|
| Volume | Rock avalanche | Large rockfalls | Medium rockfalls | Small rockfalls | Single Blocks |
| Distinction | Total volume > 1,000,000 $m^3$ | Total volume close to 1,000,000 $m^3$ | Total volume close to 10,000 $m^3$ | Total volume approximately 100 $m^3$ or fragments with more than 200 mm diameter | One or few single fragments < 200 mm diameter |

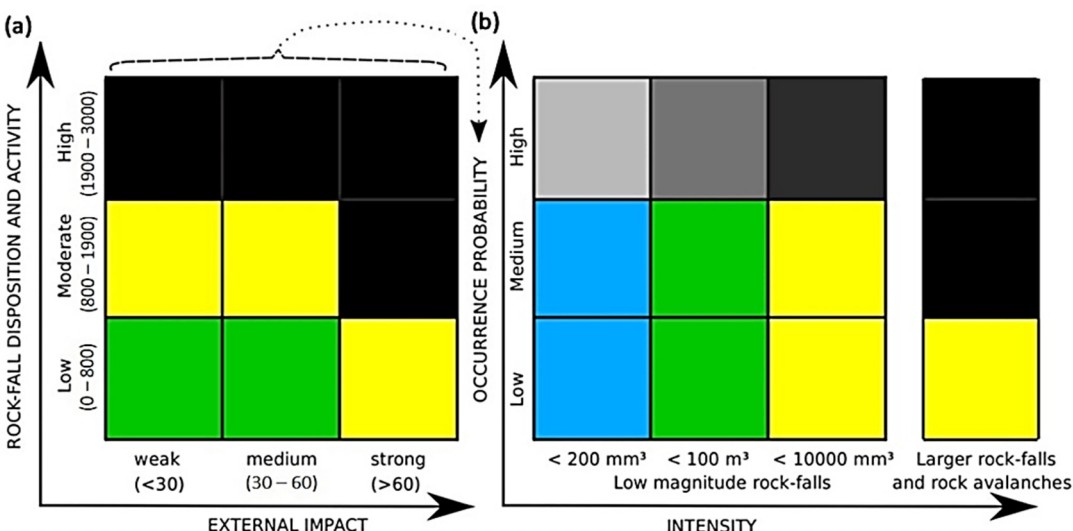

**Figure 2.** (**a**) Probability Matrix: green, "low"; yellow, "medium"; and black, "high". (**b**) Hazard Matrix: blue, very low hazard; green, low; yellow, medium; gray (with grayscale), high, modified from [8].

### 3.2. The Proposed Method

The new proposal kept the parameters related to rock mechanics and signs of activity because they are general and applicable to a great variety of situations. The intensity was also kept because it is a measure of block volume, which is related to the energy of the block. Quantifying the energy of a falling block is not an easy task; hence, by considering the block volume it is possible to evaluate this effect indirectly, depicting the intensity of the event.

The first adaptation proposed consisted of the substitution of geomechanical environmental parameters by the geometry and characteristics of the slope and the catchment area, which are important issues regarding the block trajectory in falling movements. In the Bavarian Alps, these parameters were not considered; all the situations were assumed previously hazardous because of the cliff geometry. Anyway, the catchment area is not a concern in the region of Bavarian Alps.

The deformations and mass movement in the slope toe, which were evaluated by the original method in geomechanical environment disposition, are important for high slopes or cliffs. They are not a concern for the slopes analyzed in this research. The geometry of the slope and the catchment area are much more important than those parameters in the context of the slopes analyzed, because failure mechanisms in the studied slopes are not deformation related.

In this adaptation, the type of basement was replaced by the slope height, a crucial parameter, according to [18]. The large-scale deformations item was replaced by the slope dip and surface. Finally, the mass movement in the slope toe was replaced by the characteristics of the catchment area.

The type of basement was a critical parameter in the geomechanical environment item of the original method; thus it received the highest score. In the proposed methodology, the slope height was considered crucial, because of it being directly related to the block trajectory and energy. Furthermore, when it is not possible to build catchment areas or barriers, i.e., in operational open pit slopes, one of the measures is to decrease the height of the slope by building benches.

The score of the slope height is given by Equation (1), where *ISH* is the score and *H* is the slope height (m). This Equation was presented by [19]; it was adapted from [20], based on RHRS.

$$ISH = e^{0.07996H} \tag{1}$$

The slope dip and the roughness-waviness of the slope surface can modify the trajectory of the block (Table 3). The most unfavorable situation is a rough slope with a dip between 30° and almost 75° because, in this range of inclination, the block can roll or jump [21]. Moreover, the block can collide with the irregularities of the slope face and bounce highly and randomly. An intermediary situation is a vertical slope with roughness and overhangs, because a block would behave like a projectile. The favorable situation is a vertical slope, without roughness, like benches, scars or overhangs.

**Table 3.** Modified scores from [8].

| Slope Dip and Surface | Rough SlopeDip 30°–75° | Vertical Slope with Roughness and Overhangs | Vertical Slope, without Roughness and Overhangs |
|---|---|---|---|
| Score | 200 | 100 | 0 |
| Catchment area | No area; or area with low distance and high inclination. No vegetation. | Moderate distance and inclination. Low or no vegetation. | Larger distance, plane. Or an adequately designed catchment area, like in highways. |
| Score | 200 | 100 | 0 |
| External impact | Unfavorable | Fair | Favorable |
| Precipitation range (mm/days) | >129/6 | 40–129/6 | <40/6 |
| Score | 50 | 20 | 0 |
| Seismic effects | High seismic zone or damage due to blasting in mines | karst regions, blasting in mines or heavy vehicles and/or medium seismic zone traffic | No seismic effects |
| Score | 50 | 20 | 0 |

If there are vulnerable elements at the slope base, the hazard is directly related to the block path and the available catchment area. If the catchment area is large and approximately flat, the chance of a block reaching the exposed element is low. Another relevant element is vegetation because it can act as a natural barrier, absorbing the block energy (Table 3).

In RHRS, the catchment area is a projected structure with a width and a depth designed according to the height and the slope dip, using the Ritchie abacus [21]. In the new proposal, another approach regarding the catchment area was considered, in order to represent the hazard for urban or mine slopes. Therefore, the catchment area in this proposal is related to the distance between the slope and the exposed element. The inclination, presence of vegetation, talus deposit and the measured distance in relation to the block dimensions are observed (i.e., a block with 6 m of length in an inclined area without vegetation, with around 8 m of distance, is considered an unfavorable situation).

Finally, regarding the weights attributed to the slope dip and the catchment area, for the sake of flexibility, an intermediary situation was proposed between unfavorable and favorable situations.

The second adaptation proposed in this research regards the external impact. The item was kept, but the precipitation values were adapted to cover a variation in precipitation more suitable for rainy regions than the one in which the original method was developed. The earthquake evaluation was replaced by the evaluation of seismic effects, including those related to blasting and traffic. In the area of study, earthquakes are not a concern.

Regarding the external impact due to precipitation, ref. [22] studied the relationship between mass movements and rainfall in the region of Mariana and Ouro Preto; after several statistical analyses, ref. [22] concluded that the accumulated precipitation of 48.2 mm in six consecutive days triggers slope failures. According to the same author, the critical precipitation amount is 129 mm/6 consecutive days, and in rainy seasons (October to March) the critical precipitation is often reached. Ref. [22] analyzed the historical series of

precipitation in the region; thus, the author could establish the maximum and minimum precipitation values, which were used for the proposed approach (Table 3). In Table 4 the parameters surveyed in field are presented.

**Table 4.** Parameters surveyed in field.

| Parameters | | Unit | Measurement/Classification |
|---|---|---|---|
| Uniaxial Compressive Strength (UCS) | | MPa | Geological Hammer/ ISRM Classification [17] |
| Weathering Degree | | - | ISRM Classification [17] |
| Discontinuities | Orientation | Dip/Dip Direction | Brunton Compass |
| | Spacing | Metric System | Tape |
| | Length | Metric System | Tape |
| | Roughness | - | Barton [23] |
| | Aperture | Metric System | Tape |
| Slope and catchment area | Height and inclination | Metric System | Tape and Laser Tape |
| Block dimension | | Metric System | Tape |
| Vegetation | | - | - |

## 4. Results

### 4.1. Characterization of Rock Masses—Slope-1

Slope-1 (Cabanas neighborhood) consists of a quartzite of the Itacolomi Group. Two areas (PR1 and PR2) were selected for the study due to their geomechanical and structural conditions. These areas have discontinuity planes along the entire slope surface; however, the direct measurement of the parameters at the field is limited to accessible points. These areas are prone to rockfalls, which can reach the buildings below the slope (Figure 3).

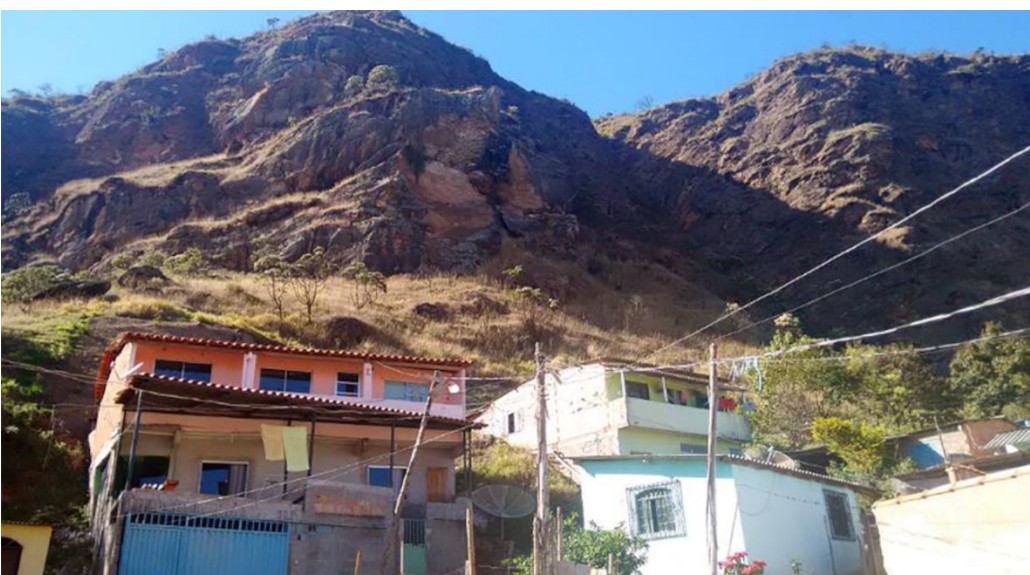

**Figure 3.** Slope-1 overview.

Both areas PR1 and PR2 are very large, 120 m and 75 m in length, respectively. Thus, the geometry of the slope and the characteristics of the catchment area vary considerably along the slope length. Hence, these areas have been split into homogeneous sectors. Variables that define these sectors are the slope height (H), the slope dip ($\psi$s) and the characteristics of the catchment area (Table 5).

**Table 5.** Geometry of the slope for each sector in Slope-1.

| Sector | H (m) | $\psi_s$ (°) | Length (m) |
|---|---|---|---|
| S1A | 55 | 75 | 60 |
| S1B | 30 | 68 | 25 |
| S1C | 29 | 57 | 35 |
| S2A | 80 | 60 from the top to 34.5 m<br>69 from 34.5 m to the base | 30 |
| S2B | 59 | 72 | 17 |
| S2C | 78 | 73 | 28 |

The areas and sectors are shown in Figure 4. In Table 6, the geometry of the catchment areas for each sector is presented, i.e., its inclination, $\psi_{ca}$ and its distance from buildings, D. Table 6 also describes the general characteristics of the catchment areas like the presence or absence of vegetation and the occurrence of rock debris or rock blocks.

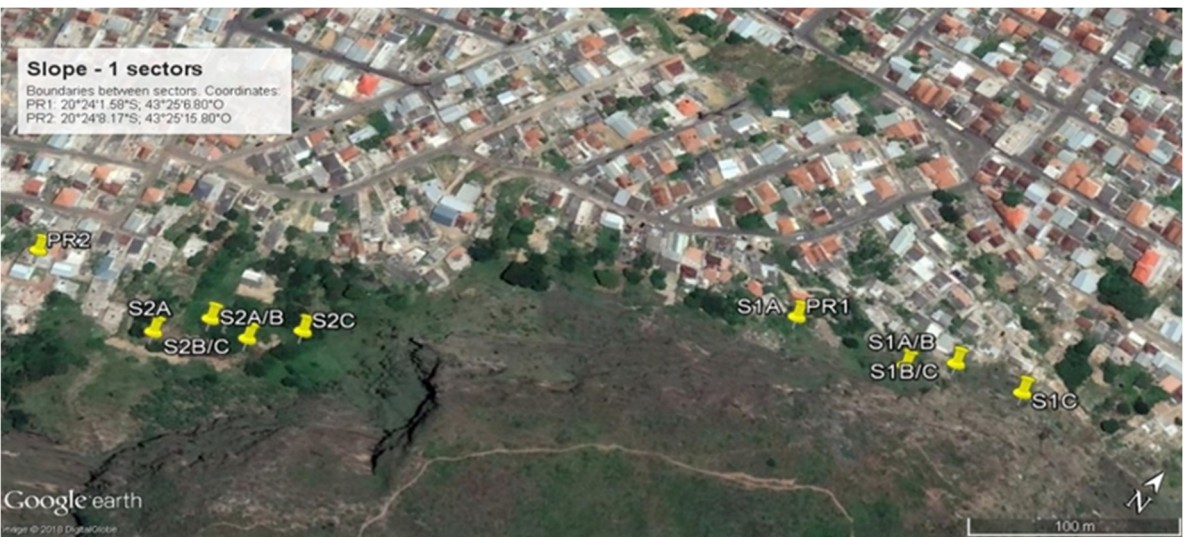

**Figure 4.** Areas (PR1 and PR2) and sectors of the slope in Cabanas (Google Earth, accessed on 16 December 2016).

**Table 6.** Catchment area features for each sector in Slope-1.

| Sector | $\psi_{ca}$ (°) | D (m) | Characteristics |
|---|---|---|---|
| 1A | 18 | 24 | Soil and grass. |
| 1B | 15 | 17 | Soil, grass and some blocks. |
| 1C | 41 | 9 | Soil, debris and blocks. |
| 2A | 25 (talus cover); 13 (pasture). | 20 (talus cover); 18 (pasture). | Talus cover: small trees, soil, debris and blocks.<br>Pasture: soil and grass. |
| 2B | 25 | 45 | Soil and grass |
| 2C | 11 | 50 | Soil and grass |

Three discontinuity sets were observed in both areas (Table 7). Kinematic conditions for wedge failures were found in both areas. Set 1, developed through foliation, is very persistent in both areas, compared to other sets. Its roughness and persistence are also distinctive, due to its planar surface in the slope face (Table 8). Filled discontinuities for all sectors have a soft sand filling. Weathering degree is low for all sectors; the rock mass is

fresh or slightly weathered (Table 8). Uniaxial compressive rock strength is relatively high; it varies from strong (50 to 100 MPa) to very strong (100 to 250 MPa).

**Table 7.** Geometric characteristics of the discontinuity sets of PR1 and PR2 areas in Slope-1.

| Area | Set | Dip/Dip Direction (°) | Spacing (m) | Trace Length (m) | Roughness |
|------|-----|----------------------|-------------|------------------|-----------|
| PR1 | Set 1 (foliation) | 21/139 | 0.42 | 20 | Rough, planar |
| | Set 2 | 60/338 | 1.43 | 3 | Rough, undulating |
| | Set 3 | 62/038 | 0.76 | 3 | Rough, undulating |
| | Slope Face | 67/317 | - | - | |
| PR2 | Set 1 (foliation) | 27/117 | 0.24 | 20 | Rough, planar |
| | Set 2 | 82/339 | 0.44 | 3 | Rough, undulating |
| | Set 3 | 55/280 | 0.76 | 3 | Rough, undulating |
| | Slope Face | 70/318 | - | - | |

**Table 8.** Characteristics of the sets and rock mass for each sector in Slope-1.

| Sector | Aperture (cm) | Filling | Weathering (ISRM 1981) | Strength (ISRM 1981) |
|--------|---------------|---------|------------------------|----------------------|
| S1A | Closed | None | W1 | R5 |
| S1B | 1–5 | Soft sand | W2 | R4 |
| S1C | 1–10 | Soft sand | W2 | R4 |
| S2A | Closed | None | W1 | R5 |
| S2B | Closed | None | W1 | R5 |
| S2C | 1–5 | Soft sand | W1 | R4 |

*4.2. Characterization of Rock Masses—Slope-2*

Slope-2 (railway between Ouro Preto and Mariana) consists of schist of the Sabará Group. This slope is homogeneous along its extent of 35 m. During the railway construction, slopes were cut on each one of the two sides of the railway, named A (Figure 5, left side) and B (Figure 5, right side). Slope A is 6 m high, and Slope B is 18 m high. Only Slope B has a catchment area, with 0.80 m of width and 0.40 m of depth (Figure 6). This catchment area was not designed to catch the falling blocks; it is related to the amount of ballast used to build the structure. In the images of the slopes, a level staff (4 m) was used as scale.

Three discontinuity sets were identified in Slopes A and B. In Table 9, the dip and the dip direction of the sets and slopes, the spacing, the trace length and the roughness are shown. Kinematic conditions for wedge failures were found for Faces A and B; block toppling can also be a concern in both faces. All the discontinuities are closed. The rock weathering degree is W2, and the uniaxial compressive strength is R3/R4, i.e., 25–50 MPa/50–100 MPa, which suggests a moderate to hard rock mass.

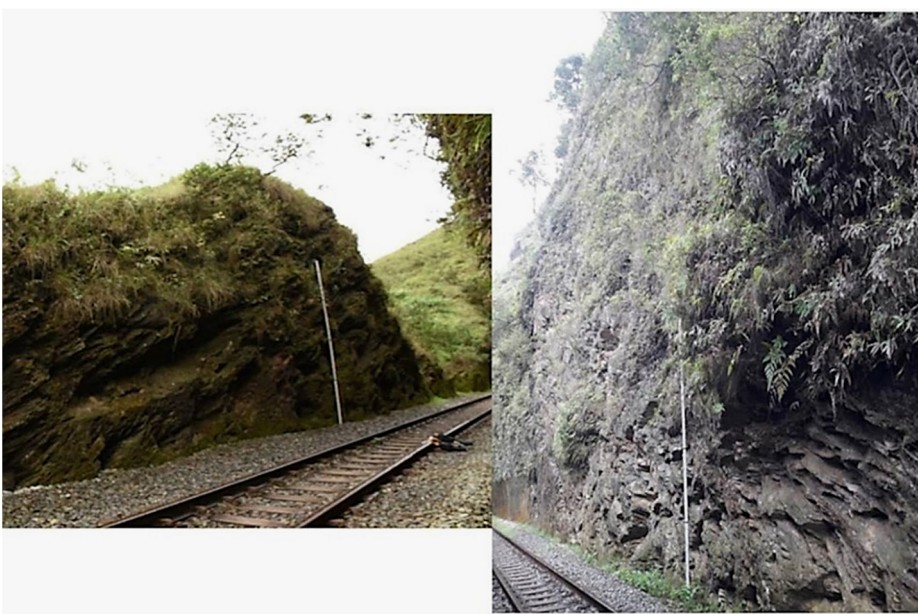

**Figure 5.** Railway showing sides A and B of Slope-2.

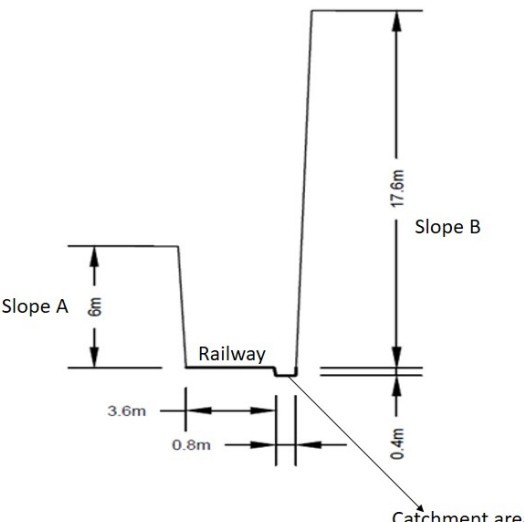

**Figure 6.** Sketch of the railway slopes.

**Table 9.** Characteristics of the discontinuity sets in Slope-2 [24].

| Set | Dip/Dip Direction (°) | Spacing (m) | Trace Length (m) | Roughness |
|:---:|:---:|:---:|:---:|:---:|
| Set 1 | 37/194 | 0.25 | 15 | Smooth |
| Set 2 | 44/139 | 0.42 | 10 | Slightly rough |
| Set 3 | 48/040 | 0.38 | 10 | Slightly rough |
| Slope A | 87/075 | - | - | - |
| Slope B | 88/259 | - | - | - |

*4.3. Characterization of Rock Masses—Slope-3*

Slope-3 (Vila Aparecida neighborhood) consists of schist of the Sabará Group. The slope height is 6.5 m. It is homogeneous and has no catchment area; the sidewalk is just below the slope toe (Figure 7). In Figure 7 a level staff of 4 m was used as a scale for the slope.

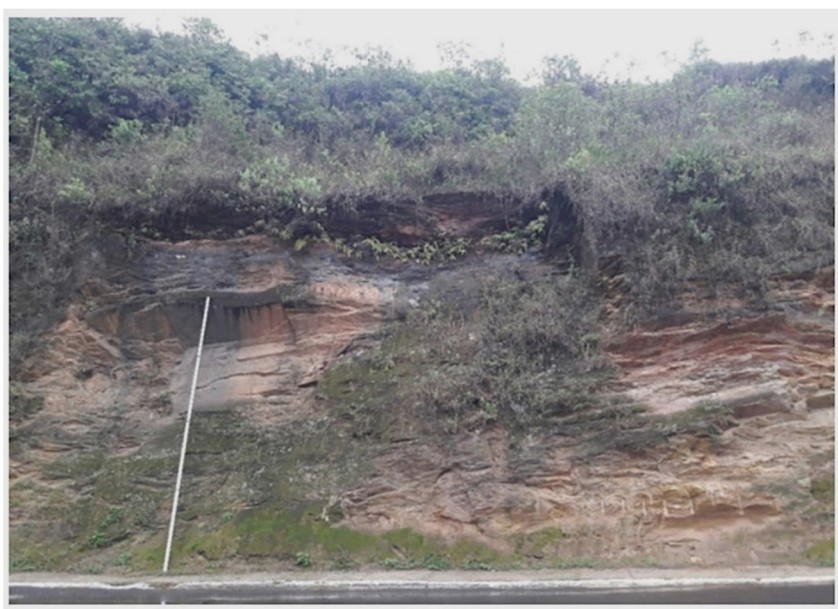

**Figure 7.** Slope-3.

Three discontinuity sets were observed in this slope. Their characteristics are presented in Table 10. All the discontinuities are closed. Kinematic conditions of planar, wedge and toppling failures were observed.

**Table 10.** Characteristics of the discontinuity sets in Slope-3 [25].

| Set | Dip/Dip Direction (°) | Spacing (m) | Trace Length (m) | Roughness |
|---|---|---|---|---|
| Set 1 | 43/211 | 0.24 | 3–10 | Slightly rough |
| Set 2 | 79/298 | 0.57 | 1–3 | Slightly rough |
| Set 3 | 46/044 | 0.36 | 1–3 | Slightly rough |
| Slope | 76/040 | - | - | - |

The rock is highly weathered (W4) and very weak, with uniaxial compressive strength in the range 1 to 5 MPa (R1), both conditions classified according to the ISRM (1981) notation.

*4.4. Rockfall Hazard Assessment*

Rockfall hazard was assessed in the three slopes, according to the Bauer & Neumann [8] method and the new proposal. In both methods, the maximum and minimum rainfalls were considered, in order to assess the hazard in rainy and dry periods.

4.4.1. Bauer & Neumann Original Method

The disposition, activity and external impact scores were obtained according to Table 1; the susceptibility results are presented in Table 11. The rockfall hazard quantification is obtained by the use of the matrix presented in Figure 2b; it is presented in Table 11.

Despite the differences observed in the field, all the slopes presented medium rockfall probability and low hazard, according to the Bauer & Neumann [8] method.

The susceptibility scores of the sectors in Slope-1 are close. All the sectors were classified as low rockfall hazard slopes (Table 11). This result was predictable because the geometry of the slope and the catchment area do not influence the scores in the original method. However, by direct observation of in situ behavior, there are noticeable differences between sectors; hence the original method is not sensitive enough to the actual rockfall hazard on this slope.

**Table 11.** Rockfall susceptibility, probability and hazard of each slope according to the Bauer & Neumann [8] original method.

| Slope | Susceptibility | Maximum/Minimum Rainfall | Seismic Effects | Maximum/Minimum Rockfall Probability | Intensity | Hazard |
|---|---|---|---|---|---|---|
| S1A | 1400 | 50/0 | 0 | Medium/Medium | Up to 200 mm | Low |
| S1B | 1600 | 50/0 | 0 | Medium/Medium | Up to 200 mm | Low |
| S1C | 1600 | 50/0 | 0 | Medium/Medium | Up to 200 mm | Low |
| S2A | 1400 | 50/0 | 0 | Medium/Medium | Up to 200 mm | Low |
| S2B | 1400 | 50/0 | 0 | Medium/Medium | Up to 200 mm | Low |
| S2C | 1600 | 50/0 | 0 | Medium/Medium | Up to 200 mm | Low |
| Slope-2 A | 900 | 50/0 | 0 | Medium/Medium | Up to 200 mm | Low |
| Slope-2 B | 900 | 50/0 | 0 | Medium/Medium | Up to 200 mm | Low |
| Slope-3 | 1700 | 50/0 | 0 | Medium/Medium | Up to 200 mm | Low |

The results for Slope-2 are coherent with observations in the field. This slope is stable; no blocks or debris have been observed near the railway trail; no scars suggesting block detachments of the slope were identified in the field.

Slope-3, the one with the worst conditions, has the same classification of the Slope-1 sectors. Once again, the original method is not sensitive to rockfall hazard, when considering field conditions of these slopes, which are very different from those analyzed by Bauer & Neumann [8].

High rainfall can lead to geotechnical problems in Slope-3. Sliding and toppling have been recurrent failures in this slope during rainy seasons.

4.4.2. The New Proposal

The results obtained by the application of the proposal presented in this research are showed in Table 12.

**Table 12.** Rockfall susceptibility, probability and hazard of each slope according to the proposed method.

| Slope | Susceptibility | Maximum/Minimum Rainfall | Seismic Effects | Maximum/Minimum RockfallProbability | Intensity | Maximum/Minimum Hazard |
|---|---|---|---|---|---|---|
| S1A | 1781 | 50/0 | 0 | Medium/Medium | Up to 200 mm | Low |
| S1B | 2011 | 50/0 | 0 | High/High | Up to 200 mm | High |
| S1C | 2010 | 50/0 | 0 | High/High | Up to 200 mm | High |
| S2A | 2200 | 50/0 | 0 | High/High | Up to 200 mm | High |
| S2B | 1719 | 50/0 | 0 | Medium/Medium | Up to 200 mm | Low |
| S2C | 2311 | 50/0 | 0 | High/High | Up to 200 mm | High |
| Slope-2 A | 1102 | 50/0 | 20 | High/Medium | Up to 200 mm | High/Very Low |
| Slope-2 B | 1104 | 50/0 | 20 | High/Medium | Up to 200 mm | High/Very Low |
| Slope-3 | 1902 | 50/0 | 0 | High/High | Up to 200 mm | High |

Slopes S1B, S1C, S2A, S2C and Slope-3 have a high probability of rockfall occurrence and were classified as high hazard (Table 12). The slopes more susceptible to rockfalls were S2A and S2C due to their height. However, in these sectors, the catchment areas are far from the buildings, contributing to the safety of these areas.

The most problematic slopes were S1B, S1C and Slope-3 due to the high hazard scores and the bad conditions of the catchment area (low distance from the buildings and absence of vegetation), which increase significantly their rockfall potential. Consequently, they require urgent intervention. The other sectors of Slope-1 require monitoring and hazard mitigation.

Slopes 2-A and 2-B have small susceptibility values. However, the rockfall probability for both slopes varies from medium to high because of the consideration of seismic effects due to traffic. These slopes ended up classifying as very low to high hazard. Therefore, the seismic effects should be analyzed deeply because there are no signs of instability in this area, such as the presence of scars or loose blocks.

## 5. Discussion

### 5.1. Comparison between Methods and General Comments

The new proposal showed an increase in the susceptibility scores for all the slopes (Table 12), which, in the majority of cases, combined with the external impact (Table 12, Figure 1a) resulted in hazard classification changes (Table 12, Figure 1). This is because the new proposal is more sensitive to parameters, such as: the slope geometry and characteristics of the catchment area. In the original method, instead of analyzing these parameters, the geomechanical environment is analyzed, which does not show any changes among the analyzed slopes, being an irrelevant item for the types of slopes in which this research is focused.

In the original methodology [8], the main problem is related to the geomechanical environment item, because the parameter "type of basement" considered by the authors seems to focus on specific issues concerning the Bavarian Alps. In addition, it considers a wide variety of rock masses as a favorable situation under the label of "other formations". Furthermore, it includes deformation parameters that do not play an important role in the slopes analyzed. The new proposal solves this problem by changing the type of basement by the slope height.

In the new proposal, the impact of maximum rainfall and seismic effects is quite significant. For instance, the same slope is classified as a very low rockfall hazard, when the score for "external impact" is minimal and as a high rockfall hazard, when the score for "external impact" is maximal. This issue appears in Slope-2, which has good structural conditions and no indications of activity. However, considering that this slope is exposed to the vibrations produced by traffic flow, the hazard score changed completely. In this case, the external influences can have more impact on the rockfall hazard than the susceptibility, which is controversial in this particular situation. More research on these slopes would be necessary to evaluate these effects.

Finally, the scores attributed to the characteristics of the slope and the catchment area presented coherent results because the higher slopes had a considerably high susceptibility score. In addition, by considering the slope dip, the surface roughness and the catchment area characteristics, more flexibility in the analyses is introduced.

For the new proposal, the slopes S1C, S2A, S2C and Slope-3 presented high hazard scores. The slope height of S2A and S2C in the Cabanas neighborhood was the most influencing parameter in their high hazard scores. In S1C and Slope-3, the rock mass and the bad conditions of the catchment area influenced their scores. The weathering of the rock mass in Slope-3 is the worst condition.

It is important to observe that the main objective of this work was to propose a new easy-to-use and inexpensive rockfall hazard system, suitable for slopes in several contexts (like urban environments, railways, mines, etc.). The system proposed was based mainly in a pre-existent method proposed by [8]. However, parameters considered in other methodologies of rockfall hazard were incorporated in the proposal, such as the RHRS [10], and the classification methods, such as RMR [16] and SMR [15].

There are other rockfall hazard approaches based on the estimation of rockfall trajectories [26]; other systems consider the frequency of rockfalls or accidents due to rockfalls [27].

The problems of these approaches are the dependence of simulation paths or monitoring data; these are not always available. The system proposed is independent of these issues. Moreover, it indicates the regions that the monitoring and more detailed studies are recommendable.

As mentioned previously, the majority of the empirical methods to determine the rockfall hazard are limited to a single context, i.e., some of them are suitable to urban and alpine rock slopes, for example, the methods proposed by [8] and [13], others only to highways. The results shown in this work proved that appropriate methods for urban alpine slopes are not sensitive and representative for slopes in other contexts, such as the mining industry. The same observation can be extended to methodologies developed for highway slopes, such as the RHRS [10].

The system proposed is flexible to several contexts and can be applied in any type of rock slope. The method was applied in urban and railway rock slopes with consistent results.

The insertion of seismic effects due to blasting was included in the proposal to permit its application to mine slopes. Many previous proposals regarding rockfall assessment are not suitable for mine slopes because they include specific variables only associated to highway slopes or urban environments.

It is important to emphasize that the rockfall is a critical event in mine slopes. This is because, in operational rock slopes, the rock mass is often disturbed by blasting and this can generate overhang blocks. Moreover, it is not common to apply reinforcement in operational mine slopes; therefore it is necessary to know the hazard level of rockfalls to remove these blocks from mine slopes.

### 5.2. Validation of the Proposed Method

Slopes of a quartzite mine, located in São Thomé das Letras town (Brazil) were chosen in order to validate the proposed method. Its application to mine slopes constitutes a challenge and could indicate if the method can be used in other types of slopes. The mine was selected because it presents slopes prone to rockfalls. Two slopes were chosen for validation; one of them had its access prohibited due to the instability caused by rockfalls (Slope X) and the other presented good structural conditions and no activity indications, being considered stable (Slope Y). Slope X and Slope Y are shown in Figures 8 and 9, respectively.

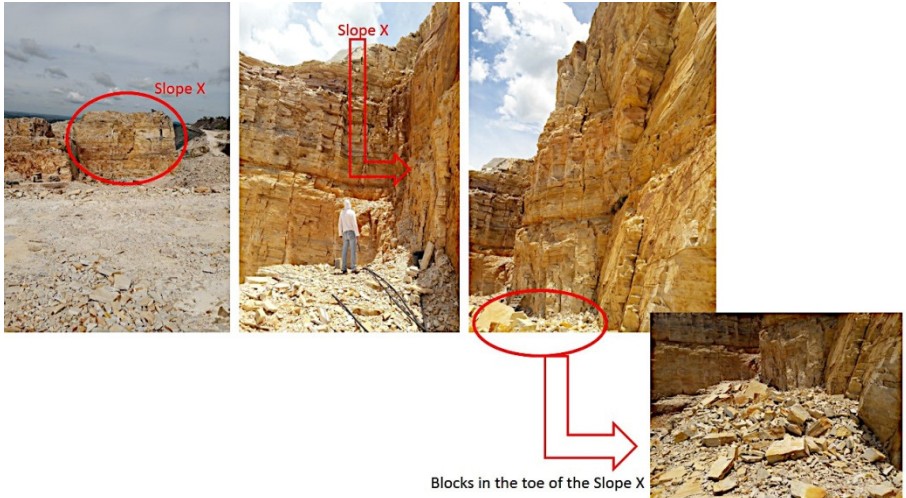

**Figure 8.** Slope X.

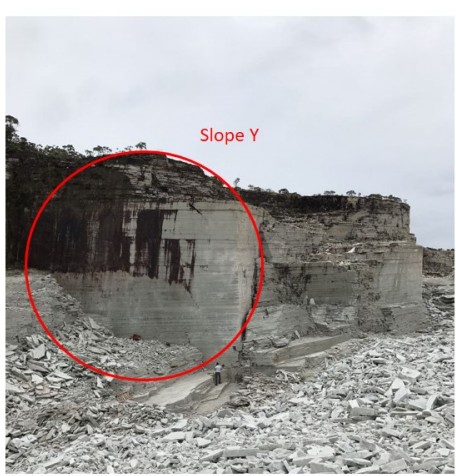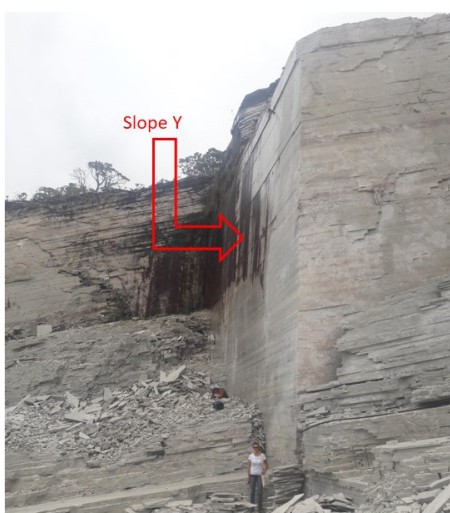

**Figure 9.** Slope Y.

Both slopes have rock masses with the same weathering degree condition and intact uniaxial compressive strength degree. The rock masses are fresh to slightly weathered (W1 or W2) and very hard (R5). The main discontinuity set of the rock masses is the foliation, whose average orientation is 11/240 (dip/dip direction), spacing ranging from 3 cm to 30 cm and persistence in the rock mass scale, i.e., more than 10 m when the length of the slope is largest than 10 m.

Slope Y has only the foliation discontinuity set. Slope X has three discontinuity sets: the foliation and two joints with average orientations equal to 64/120 and 76/065. The characteristics of the discontinuity sets are the same for both slopes. The foliation has a planar and smooth surface, and the joints have planar and rough surfaces. The aperture of the discontinuity sets are in the range of 1 mm to 1 cm, and the cracks have apertures around 5 to 10 cm. These cracks were caused by blasting. Sliding planes are noticeable in Slope X.

Slope X and Slope Y have heights equal to 16.78 m and 10 m, respectively. Both slopes are vertical. Slope X presents several indications of activity, such as failed blocks at the slope toe and several scars and cracks at the slope surface. Slope 2 presents a regular surface with no activity indications.

Failed blocks of Slope X are of low magnitude and small-scale, i.e., it has a total volume of about 100 m$^3$ and/or fragments with a diameter above 200 mm. The slopes did not present a catchment area.

The rockfall hazard conditions of Slopes X and Y were defined according to the method proposed in this research. In the analyses, the maximum and minimum rainfalls were considered. The seismic effects typical of mine slopes are related to blasting and heavy vehicles and/or medium seismic zone traffic. Table 13 shows the susceptibility scores, the maximum and minimum rockfall probabilities and the hazard conditions of Slopes X and Y.

**Table 13.** Rockfall susceptibility, probability and hazard of Slope X and Slope Y according to the proposed method.

| Slope | Susceptibility | Maximum/Minimum Rainfall | Seismic Effects | Maximum/Minimum Rockfall Probability | Intensity | Hazard |
|---|---|---|---|---|---|---|
| X | 2003.8 | 50/0 | 20 | High/High | Up to 200 mm | High |
| Y | 702.2 | 50/0 | 20 | Low/Low | Up to 200 mm | Low |

The results obtained for both slopes were consistent, considering the slope characteristics observed in the field. The structural condition of the rock masses and the signs of

activity were the most relevant parameters in the hazard analyses. Slope X presented high susceptibility and a high probability of rockfall and high hazard, considering maximum and minimum rainfall. Slope Y presented low susceptibility and a low probability of rockfall and low hazard for both rainfall conditions.

## 6. Conclusions

The method for rockfall hazard assessment proposed in this article is a step towards a new comprehensive methodology for rockfall risk assessment in rock slopes.

Strong points of the proposed method can be highlighted:

- The proposal is an easy to use and inexpensive method to evaluate rockfall hazards;
- Slope height, dip and the catchment area are incorporated in the proposal, which are parameters directly related to block path in a falling movement;
- Seismic effects are included in the proposal to evaluate the influence of traffic and blasting in rockfall assessment.
- Important results obtained by the method application are:
- The proposed method was applied successfully to a wide range of slope types, like urban, railway and mine slopes;
- The method proved to be efficient to quantify the most important factors affecting rockfalls.

The majority of current methods for rockfall hazard assessment are not suitable for application in mine slopes. Validation of the proposal in mine slopes showed consistence with the situation observed in the field regarding the tested slopes.

**Author Contributions:** Conceptualization, M.S.L. and P.A.-H.; methodology, L.R.C.S. and P.A.-H.; validation, L.R.C.S., P.A.-H. and T.B.d.S.; formal analysis, L.R.C.S.; investigation, L.R.C.S.; data curation, L.R.C.S.; writing—original draft preparation, T.B.d.S. and P.A.-H.; writing—review and editing, L.R.C.S., M.S.L., P.A.-H. and T.B.d.S.; supervision, M.S.L.; project administration, M.S.L. All authors have read and agreed to the published version of the manuscript.

**Funding:** This research received no external funding.

**Data Availability Statement:** Not applicable.

**Acknowledgments:** The authors acknowledge CNPq (National Counsel of Technological and Scientific Development), Fapemig (Foundation for Research Support of Minas Gerais) and the Brazilian Federal Agency for Support and Evaluation of Graduate Education (CAPES) for supporting this work and the mining company for the opportunity of the study.

**Conflicts of Interest:** The authors declare no conflict of interest.

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
