# Peer review of "A New Methodology for Rockfall Hazard Assessment in Rocky Slopes"

_mining, doi:10.3390/mining2040044_

Round 1

Reviewer 1 Report

1.      Problem statement is missing in the abstract. Why the authors wants to develop the new methodology.

2.      New Methodology applications and limitations. Comparing with the original methodology.

3.      In the introductions section, the authors must highlight the issue rock fall hazard. A few recent major issue. Some preventive approaches and the gradually switch to assessment.

4.      After discussing the assessment approaches, the authors must include the problem statement and then the methodology.

5.      The paper must reflect the study novelty.

6.      Using the journal guidelines, a separate chapter will be appropriate for the methods and materials. I mean, project details must be in the materials and method section.

7.      Photographs of the slopes?

8.      State of the art?

9.      The authors should avoid the frequently use of “this article”.

10.   Use the journal guidelines for the sentence starting with reference.

11.    Define the symbols and abbreviation for the first time in the text.

12.   Include the Comparison of the proposed methodology with existing approaches.

13.   Compared to the exiting approaches, what is the level of practicality of the revised approach.

14.   Try to avoid the referencing in the abstract and conclusion.

15.   Show in a tabulated form, the parameters along with their rating for the original and new methods.

16.   The conclusion must be more concise and must reflect the study. No explanation is required at this stage of the manuscript.

Author Response

Reviewer 1:

  1. Problem statement is missing in the abstract. Why the authors wants to develop the new methodology.

      The original methodology is quite restricted, as it was proposed for application in the Bavarian Alps. However, the idea presented is interesting; so we decided to adapt it to more flexible situations. So, we rewrote part of the abstract to explain the objective of the article:

      “The original method is appropriate to high alpine rocky slopes exposed to large scale deformations. It evaluates the parameters related to the geomechanical characterization of rock mass, indications of activity, external influences and event intensity. The original methodology was modified to consider different contexts, including geological, climatic and social environments.”

  1. New Methodology applications and limitations. Comparing with the original methodology.

      Text added in topic 2. Rockfall hazard and risk assessment methodologies:

“The methodologies aforementioned are efficient, and some of them are widespread internationally, such as RHRS [12]. Furthermore, they are easy to use, which is an important feature in geotechnical hazard and risk analysis routines. However, they present some limitations regarding applicability. Some of them are suitable only for highway slopes, others only for urban slopes, precisely for alpine regions. Others do not present a hazard analysis. Thus, it is important to propose an easy-to-use methodology, as well as those cited, but more flexible and able to be applied in different contexts. Thus, the described methods were used as a basis for this proposal, considering their strong points and disregarding their weaknesses.”

Text added in the conclusion:

“It increased flexibility to the original method proposed initially to high alpine rock slopes, and enables the application of the methodology to a wide range of slope types.”

As explained in the paper, the methodology a step towards a new comprehensive methodology for rockfall risk assessment in rock slopes. Thus, it is open to implement a risk analysis by assessing the consequences of rockfall. Another work will be soon available in the future to propose a complete risk analysis for rockfall.

  1. In the introductions section, the authors must highlight the issue rock fall hazard. A few recent major issue. Some preventive approaches and the gradually switch to assessment.

      Text added to the Introduction:

      “This condition worsens when monitoring measures are not available or when the rockfall hazard is neglected. Monitoring measures are not available especially in peripheral urban areas and, sometimes, in ecological or adventure tourism areas. A recent example of a serious accident involving a high magnitude rockfall occurred in January, 2022 in Capitólio city, a cliff region of Minas Gerais State of Brazil. In this case, a high quartzite rock block toppled and hit a boat with tourists in a lake located in this region [8]. Another accident involving rockfall, also in Brazil, occurred in January, 2021 at a quarry in the metropolitan region of Salvador, State of Bahia, when a rock block fell from an operational slope onto an excavator, causing the death of one operator [9].

In populated mountainous regions, rockfalls constitute a major hazard, once they can cause damage to properties and personal injuries. Therefore, a geotechnical hazard assessment in these areas is essential, and it consists of the first step of future mitigation planning and risk management. Through geotechnical hazard assessment it is possible to define the areas with the most urgency of intervention and with the need of control or mitigation measures. These measures can include a constant monitoring plan, block support with bolts and high-resistance screens and removal of overhanging blocks.

In view of the importance of rockfall hazard classification, the main objective of this paper is to propose an easy-to-use approach to rockfall hazard assessment. This approach was adapted from the methodology proposed by [10]. This methodology is easy to use, but it is only suitable for high urban slopes, in alpine regions. Thus, the methodology proposed in this research aims to improve the original proposal by the accurate description of the slope geometry, adaptation of the methodology to rainy regions and inclusion of other seismic situations, like mine blasting and heavy equipment traffic.”

  1. After discussing the assessment approaches, the authors must include the problem statement and then the methodology.

      Text added:

“Three slopes susceptible to rockfalls were selected to carry out the rockfall hazard analysis. In case of rockfall occurrence in these slopes, serious consequences are expected, with material losses and damage to the physical integrity of the exposed people. These slopes are: Slope-1, located in the Cabanas neighborhood (Mariana, Minas Gerais District, Brazil); Slope-2, located on the railway which connects the towns of Ouro Preto and Mariana and Slope-3, located in the Vila Aparecida neighborhood (Ouro Preto, Minas Gerais District, Brazil).

“For all slopes, field inspections were made with the purpose of reconnaissance of the area and delimitation of access points to carry out a survey of geotechnical and local characteristics. It was possible to observe and survey both the parameters related to the original methodology developed by [10], as well as other parameters that apparently had an influence on the rockfall hazard in these places, but which were not taken into account in the original methodology of [10]. These steps made possible the proposition of a new system of rockfall hazard analysis, based on the proposal of [10]. In the following topics, the original methodology developed by [10] is presented, as well as the new proposal. Field observations and the application of both methods are presented in the results topic.”     

  1. The paper must reflect the study novelty.

      Text added in the discussion topic:

      “It is important to observe that the main objective of this work was to propose a new easy-to-use and inexpensive rockfall hazard system, suitable for slopes in several contexts (like urban environments, railways, mines). The system proposed was based mainly in a pre-existent method proposed by [10]. However, parameters considered in other methodologies of rockfall hazard were incorporated in the proposal, as the RHRS [12], and the classification methods, as RMR [18] and SMR [17].

There are other rockfall hazard approaches based on estimation of rockfall trajectories [28]; other systems consider the frequency of rockfalls or accidents due to rockfalls [29]. The problems of these approaches are the dependence of simulation paths or monitoring data; not always available. The system proposed is independent of these issues. Moreover, it indicates the regions that the monitoring and more detailed studies are recommendable. 

As mentioned previously, the majority of the empirical methods to determine the rockfall hazard are limited to a single context; i.e., some of them are suitable to urban and alpine rock slopes, for example, the methods proposed by [10] and [15]; others only to highways. The results shown in this work proved that appropriate methods for urban alpine slopes are not sensitive and representative for slopes in other contexts, as mining industry. The same observation can be extended to methodologies developed for highway slopes, such as the RHRS [12].”

  1. Using the journal guidelines, a separate chapter will be appropriate for the methods and materials. I mean, project details must be in the materials and method section.

      Ok.

  1. Photographs of the slopes?

      Ok.

  1. State of the art?

      Ok.

  1. The authors should avoid the frequently use of “this article”.

      Ok.

  1. Use the journal guidelines for the sentence starting with reference.

Ok.

  1. Define the symbols and abbreviation for the first time in the text.

Ok. 

  1. Include the Comparison of the proposed methodology with existing approaches.

      Ok.

  1. Compared to the exiting approaches, what is the level of practicality of the revised approach.

      Ok.

  1. Try to avoid the referencing in the abstract and conclusion.

Ok.  

Reviewer 2 Report

This study investigates the “Rockfall Hazard Assessment Approach for Rock Slopes”. While reviewing this paper, I focussed on the structure, content, coherence, grammar, typos, etc. I was impressed with the structure and analysis of the report, however, the abstract and the conclusion should be worked on to reflect the findings of the study. I suggest the abstract should be reviewed to reflect the scientific background and findings of the study. In addition, the conclusion should be written in a concise model.

I suggest the topic should be modified to “a new methodology for rockfall risk assessment in rock slopes”.

With regards to the raised comments, the manuscript should be accepted with minor corrections

Author Response

Reviewer 2:

This study investigates the “Rockfall Hazard Assessment Approach for Rock Slopes”. While reviewing this paper, I focussed on the structure, content, coherence, grammar, typos, etc. I was impressed with the structure and analysis of the report, however, the abstract and the conclusion should be worked on to reflect the findings of the study. I suggest the abstract should be reviewed to reflect the scientific background and findings of the study. In addition, the conclusion should be written in a concise model.

I suggest the topic should be modified to “a new methodology for rockfall risk assessment in rock slopes”.

With regards to the raised comments, the manuscript should be accepted with minor corrections

Comments:

The title has been changed: “A New Methodology for Rockfall Hazard Assessment in Rocky Slopes”

The word hazard was kept because we did not carry out a risk analysis, as we did not consider the consequences of rockfalls; only the susceptibility.

Regarding the abstract, we changed it according to the suggestions presented:

“Abstract: This article presents an approach to rockfall hazard assessment for rocky slopes based on a previously published rockfall hazard methodology. The original method is appropriate to high alpine rocky slopes exposed to large scale deformations. It evaluates the parameters related to the geomechanical characterization of rock mass, indications of activity, external influences and event intensity. The original methodology was modified to consider different contexts, including geological, climatic and social environments. Parameters related to the external influences were modified; the geometry and characteristics of the slope and the catchment area were introduced. The original methodology and the new proposal were applied to two urban slopes and one railway slope in order to test and compare the methods. The original proposal could not represent the rockfall conditions of these slopes. The new proposal was validated using two mine slopes, whose conditions of stability are known. The results of the analyses with the urban slope and the railway slope were coherent with the situation observed at the field. The validation in the mine slopes showed that this approach is applicable in several situations, being able to determine how hazardous a slope is in relation to rockfall events.”

Text added to the conclusions:

“The results generated by applying the new methodology on urban, railway and mining slopes were consistent with the situation observed in the field. Therefore, it can be considered that this methodology is effective, as it includes the most relevant parameters for a hazard rockfall analysis on any type of slope, generating representative results. Furthermore, it is easy to use and inexpensive, as it does not depend on any software or even laboratory tests.”

Reviewer 3 Report

Dear Authors,

In the following you may kindly find some comments & suggestions regarding your submission:

o   The title must be revised. It is confusing and very general. May be the study could be added!? Rock slopes or Rocky?!

o   Line 13: context could fit better than environment

o   Expand the keywords list as this is an opportunity to increase the accessibility to your article.

o   Line 28: Avoid listing so many references in a single row. 

o   Line 36: please give the surname of the author (s) before citing the source in numbers.

o   The map in Fig.1 must be graphically improved. First, you must show where Brazil is located within south America. Then, where your study areas, within Brazil. The gradient of elevation is not needed, the maps must be much simple. Please, do not forget to add, coordinates, scale bar, north sign, and legend at each map…

o   It seems that you have 5 locations in total (3+2). Please indicate them clearly on the map…

o   2.1. Concepts: there must be a reason why you have selected ISSMGE among other definitions?! Please, expand the reasons of your selection in this part.

o   I think that Section 3: Materials and Methods must include a subchapter dedicated to the study area. The information you have delivered in the introduction is not enough to introduce the study are to the reader. You must include here not only general info about the area but also some specific information about rock fall events.

o   The sketch in Fig. 6 is not understandable! Besides the dimensions, please add some keywords specifications to annotate the elements show in the sketch.

o   Fig. 6 caption is appearing twice. Please renumber!

o   I am afraid Fig. 7 and 8 are not enough to present new slopes… First, you must show them where they are located on the map in reference to the other three locations. Then the graphical material must be equivalent in scale or graphical quality. At this state this is not a good representation of the new two slopes!

o   The discussion part must be expanded discussing findings with reference to mining industry. This will bring your work closer to the aim and scope of the journal and as a result to the interest of its readership.

o   Please, cross check if all cited sources are included in the references list and vice versa! References could be more than 25 for such a study...

Thank you!

Author Response

Reviewer 3:

o   The title must be revised. It is confusing and very general. May be the study could be added!? Rock slopes or Rocky?!

 The title has been changed: “A New Methodology for Rockfall Hazard Assessment in Rocky Slopes”

o   Line 13: context could fit better than environment

In fact we mention the environment, but the context is what we wanted to refer. The original methodology is quite restricted because it was developed to apply in the Bavarian Alps.

This sentence in abstract was changed:

“The original methodology was modified to consider different contexts, including geological, climatic and social environments.”

o   Expand the keywords list as this is an opportunity to increase the accessibility to your article.

Ok:

Keywords: rockfall hazard system; probability matrix; hazard matrix; urban slopes; railway slopes, mine slopes

o   Line 28: Avoid listing so many references in a single row.

corrected

o   Line 36: please give the surname of the author (s) before citing the source in numbers.

We followed the rules of the journal.

o   The map in Fig.1 must be graphically improved. First, you must show where Brazil is located within south America. Then, where your study areas, within Brazil. The gradient of elevation is not needed, the maps must be much simple. Please, do not forget to add, coordinates, scale bar, north sign, and legend at each map…

The figure was modified

o   It seems that you have 5 locations in total (3+2). Please indicate them clearly on the map…

We have 3 locations: Ouro Preto, Mariana and São Thomé das Letras (we indicate them in the map)

o   2.1. Concepts: there must be a reason why you have selected ISSMGE among other definitions?! Please, expand the reasons of your selection in this part.

The following text was added:

“This article applies the concepts defined by the Technical Committee 32 from International Society of Soil Mechanics and Geotechnical Engineering – ISSMGE [11]. These definitions are internationally accepted and, according to [11], should be used for all zoning, reports and land use planning documents in order to avoid misunderstanding of the terms.”

o   I think that Section 3: Materials and Methods must include a subchapter dedicated to the study area. The information you have delivered in the introduction is not enough to introduce the study are to the reader. You must include here not only general info about the area but also some specific information about rock fall events.

A detailed description of the studied slopes is included in the Results section because the field survey is part of the results. The values of the parameters surveyed were used in Rockfall assessment; which is also part of the results.

o   The sketch in Fig. 6 is not understandable! Besides the dimensions, please add some keywords specifications to annotate the elements show in the sketch.

Ok.

o   Fig. 6 caption is appearing twice. Please renumber!

Ok.

o   I am afraid Fig. 7 and 8 are not enough to present new slopes… First, you must show them where they are located on the map in reference to the other three locations. Then the graphical material must be equivalent in scale or graphical quality. At this state this is not a good representation of the new two slopes!

Figures of mine slopes were modified to show details of the studied slopes.

o   The discussion part must be expanded discussing findings with reference to mining industry. This will bring your work closer to the aim and scope of the journal and as a result to the interest of its readership.

Text added to the discussion section:

“The system proposed is flexible to several contexts and can be applied in any type of rock slope. The method was applied in urban and railway rock slopes with consistent results.

The insertion of seismic effects due to blasting was included in the proposal to permit its application to mine slopes. Many previous proposals regarding rockfall assessment are not suitable for mine slopes because they include specific variables only associated to highway slopes or urban environments.

It is important to emphasize that the rockfall is a critical event in mine slopes. This is because in operational rock slopes the rock mass is often disturbed by blasting and this can generate overhang blocks. Moreover, it is not common to apply reinforcement in operational mine slopes; therefore it is necessary to know the hazard level of rockfalls to remove these blocks from mine slopes.”

As discussed above, the proposal was adapted for application in mine slopes. Besides, we use two mine slopes to validate the proposal, because systems to assess the rockfall hazard were not developed for mine slopes. However, our proposal produced consistent results when applied to mine slopes, which allows our confidence in applying it in many different contexts.

o   Please, cross check if all cited sources are included in the references list and vice versa! References could be more than 25 for such a study...

Ok.

Round 2

Reviewer 1 Report

1.      Figure must be close to the text, where it is cited.

2.      For the slope figures, the description must be enough for self-explanation.

3.      Conclusion must be concise, based on the study results and better if in bullets.

Author Response

Answer to the reviewer:

Regarding the suggestions made by the reviewer, the following points were modified:

  • the figures were moved to be closer to the text where they are cited;
  • the captions of Figures 3, 5 and 8 were modified, as the descriptions in the text are enough for explaining the figures;
  • the conclusion was modified deeply to be more concise and to present the main results of the work.

Reviewer 3 Report

Dear Authors,

Thank you for considering most of the suggestions and comments!

The final improvements could be more, especially regarding the number of cited sources, and the suggested reference map regarding Figure 8 & 9.

Yet, at this stage I think that your manuscript can be accepted.

Congratulations!

Kind regards,

Reviewer  

Author Response

Point presented by the reviewer:

"The final improvements could be more, especially regarding the number of cited sources, and the suggested reference map regarding Figure 8 & 9."

We put the location of São Thomé das Letras in the map of Figure 1.

The cited sources have been revised.